# Microbiological Quality of Red Meat Offal Produced at Australian Export Establishments

**DOI:** 10.3390/foods11193007

**Published:** 2022-09-27

**Authors:** Paul Vanderlinde, Peter Horchner, Long Huynh, Ian Jenson

**Affiliations:** 1Vanderlinde Consulting, Beenleigh Redland Bay Road, Carbrook, QLD 4130, Australia; 2Symbio Laboratories, 52 Brandl Street, Eight Mile Plains, QLD 4113, Australia; 3Meat & Livestock Australia, 40 Mount Street, North Sydney, NSW 2059, Australia; 4Centre for Food Safety and Innovation, University of Tasmania, College Road, Sandy Bay, TAS 7005, Australia

**Keywords:** offal, beef, sheep, lamb, goat, hygiene

## Abstract

A national baseline study of offal hygiene was undertaken at 17 Australian export establishments. A total of 1756 samples of different offal types were analysed for aerobic plate count (APC), generic *Escherichia coli*, and coliform bacteria. Average APC values varied from 1.51 to 5.26 Log_10_ CFU/g, depending on species and offal type. The average APC on beef, sheep, lamb, and goat offal was 3.25, 3.38, 3.70, and 2.97 Log_10_ CFU/g, respectively. There is a small but significant difference in APC on offal sampled frozen (3.26 Log_10_ CFU/g) and offal sampled fresh (3.73 Log_10_ CFU/g). *Escherichia coli* prevalence on beef, sheep, lamb, and goat offal was 15.4%, 28.1%, 17.5%, and 39.3%, respectively. The number of *E. coli* on positive offal samples ranged from 1.42 to 1.82 Log_10_ CFU/g. While the quality of some offal approach that of muscle meat, the hygienic quality of red meat offal can be understood by considering the anatomical site from which it is harvested, the usual bacterial levels found at that site, the difficulty in hygienically removing the offal from the carcase, the process prior to packing, and the chilling method used.

## 1. Introduction

Offal is an important component of some diets and is an important contributor to gaining value from an animal at Australian export meat processing establishments. Exports of beef and sheep offal from Australia have tripled over the last twenty years, with current export volumes reaching around 200,000 tonnes per annum [1]. Offal has always been considered of lower microbiological quality than meat [2]. While some studies support this assumption [3,4], others suggest that it is possible to process offal hygienically [5,6,7]. However, most of these studies looked at a limited number of samples from relatively few offal types.

It is generally accepted that the initial microbiological quality of offal is related to hygienic practices during harvest and, to a lesser extent, the nature of the offal itself. Inadequate chilling during the initial stages of storage can also be a contributing factor to poor microbiological quality [8]. Im et al. [9] showed that the microbiological quality of offal can vary widely between slaughter establishments, possibly indicating variability in hygienic processing between establishments.

A number of countries apply microbiological criteria to raw meat products, which can be interpreted as applying to both muscle meat and all types of offal. In addition to there being little to no widely accepted justification to set microbiological criteria for raw meats [10], little is known about how differences in the nature of the offal, processing steps, or other factors may influence the microbiological profile of various products. As a major exporter of meat and meat products, it is important for Australia to demonstrate to markets the hygienic quality of products produced under the Australian Export Meat Inspection System (AEMIS). It is also important to establish expected microbiological profiles for these products, as these data are generally lacking compared to muscle meats. This paper details the findings of a nationwide survey of the hygienic quality of beef, sheep, lamb, and goat offal produced at Australian export establishments and provides an interpretation of these data by considering their origin and processing methods.

## 2. Materials and Methods

### 2.1. Sample Collection

Seventeen export establishments, distributed across five out of six Australian states, took part in the survey. Ten processed beef, five processed lamb, six processed sheep, and three processed goats (some plants processed multiple species). All establishments operated under the supervision of the Australian Department of Agriculture, Fisheries, and Forestry (current name) and similar sample collection arrangements were made to those employed in previous studies [11,12]. Samples were collected between August 2018 and June 2019, with most samples collected in 2018. Generally, samples were collected on a weekly basis or when required offal types were processed. Sampling was carried out after refrigeration by establishment personnel from both frozen and chilled product, depending on how the offal type was processed at the establishment. A minimum 50 g sample was collected from the surface of frozen product in cartons. For chilled product, an individual piece of offal weighing at least 50 g was collected. All samples were individually bagged, labelled, and transported refrigerated (≤7 °C) by a commercial courier to a laboratory accredited to the ISO 17025-2005 standard [13] by the National Association of Testing Authorities, Australia. Analysis of samples commenced no later than the day following sample collection. Frozen samples that thawed during transport were not re-frozen. Offal types included in the survey were selected based on Australian export statistics and varied between species.

### 2.2. Indicator Bacteria

Samples (*n* = 1756, Table 1) were analysed for aerobic plate count (APC), *Escherichia coli*, and coliforms following the procedure detailed in AOAC 990.12 [14], AOAC 998.08 [15], and AOAC 991.14 [16], respectively. Briefly, a 25 g sample was homogenised in nine times its weight of peptone salt solution (PSS, ISO 6887-1:1999 [17]). Serial dilutions in PSS were prepared to ensure that a count was obtained for every sample where possible. Samples were plated onto appropriate Petrifilm^TM^ plates and incubated for 24 to 48 h at 35 ± 1 °C. Colonies were enumerated according to the AOAC procedure.

### 2.3. Data Analysis

For the purpose of analysis, unless otherwise stated, count data below the limit of detection (LOD) of the analytical method were assigned a value of ½ the LOD. During analysis of *E. coli* and coliform prevalence, no account was made for the sensitivity of the analytical test or of the effect of sample size on the likelihood of detection. Standard deviations were calculated using the STDEV function in Microsoft^®^ Excel^®^ (version 2208, Microsoft Corporation, Redmond WA, USA). Data were visualised in R [18] and statistical analysis (ANOVA and chi-squared test) was carried out in R or Minitab14 (Minitab Inc., State College, PA, USA) at a significance level of 0.05.

## 3. Results

Indicator bacteria were enumerated in 16 offal types collected from four species (lamb and sheep being considered separate ‘species’, segregated according to age by examining dentition): beef (*n* = 975), sheep (*n* = 160), lamb (*n* = 486), and goat (*n* = 135) processed at 17 Australian export establishments (Table 1). Sample numbers for each offal type and for each species were based on average export volumes.

A difference in the microbiological quality of offal due to source species can be estimated by examining the average prevalence and concentration of bacterial types across all offal collected. The average log_10_ APC on beef, sheep, lamb, and goat offal of 3.25, 3.38, 3.70, and 2.97 log_10_ CFU/g, respectively (Table 2). The log_10_ APC on offal from goats was significantly lower than that of offal from other species, while the log_10_ APC on lamb offal was significantly higher. Prevalence data obtained for faecal indicator bacteria (Table 2) revealed a higher prevalence of *E. coli* in goat offal with a significantly lower prevalence in beef and lamb offal. There was a lower prevalence of coliforms on beef offal compared to offal from other species (lamb and sheep data combined).

The average log_10_ APC varied between offal type within species (Figure 1), with some types having significantly different average log_10_ APC values (Table 3), though the mean counts of all the offal were similar for each species (beef 3.28 log_10_ CFU/g, sheep 3.42 log_10_ CFU/g, lamb 3.37 log_10_ CFU/g, and goat 3.34 log_10_ CFU/g). *E. coli* prevalence can vary considerably between offal types, although concentrations were low and not dissimilar (Table 4). Since coliform prevalence and concentration followed the same pattern as *E. coli* (Table 2), these results have not been further analysed.

There was a significant (*p* < 0.001) difference in the log_10_ APC on offal samples collected chilled and those collected frozen (Table 5), with counts generally being lower on frozen samples.

The difference between the APC on chilled (average, 3.73 log_10_ CFU/g) and frozen (average, 3.26 log_10_ CFU/g) samples varied with offal type. Where 30 or more samples were analysed for both chilled and frozen offal, a significant (*p* < 0.001) reduction in log_10_ APC was noted (tripe, tongue kidney, and liver). The log_10_ APC on chilled and frozen skirt samples was not significantly different (*p* > 0.05).

## 4. Discussion

The microbiological quality of offal varied between offal type and species. While the APC and prevalence of *E. coli* were higher on non-organ offal, such as tongue, tripe, and tendons, than typically found on meat, the average levels of contamination on all offal were considered acceptable. The hygienic quality of red meat offal can be understood by considering the anatomical site from which they are harvested, the usual bacterial levels found at that site, the difficulty in hygienically removing the offal from the carcase, the process prior to packing, and the chilling method used. The survey conducted here presents data that can be used to benchmark other processes.

The variability in bacterial counts on different offal types is not unexpected as they are harvested from different anatomical locations in the animal and are generally processed differently [19]. Further, some offal types are more likely to be contaminated either inherently or through cross contamination with other parts of the carcase during processing. Offal derived from organs such as the heart, kidney, and liver would be expected to be almost certainly sterile at the time of harvest, becoming contaminated during processing where they may be cross-contaminated with other products, either on the viscera table or during further processing and packaging. In the current study, organ offal (kidneys, livers, and hearts) generally had lower APCs than other offal types. Based on the amount of processing, we would expect organ offal to have a similar count to meat at the completion of processing. In the 2011 Australian national meat microbiological baseline study [11], the average APC on boneless beef was 2.22 Log_10_ CFU/g with *E. coli* found in 2.1% of samples, similar to beef kidneys, livers, and hearts in the current study. Similarly, bulk packed sheep meat in the 2011 baseline study had an APC and *E. coli* prevalence of 2.8 Log_10_ CFU/g and 12.5%, respectively [12], similar to levels found on sheep/lamb kidneys, livers, and hearts in the current study. The similarity of these data between organ offal and muscle meat indicates that they are harvested with a similar degree of attention to maintaining their hygienic quality.

Non-organ offal, such as tripe and tongue, may have intrinsically higher microbiological counts due to their function/location in the animal. Tripe can be processed in several ways [19], resulting in a large variation in microbiological quality of the final product. For example, while honeycomb tripe is often scalded with hot water prior to cooling, mountain chain tripe is not. While no attempt was made to identify the type of tripe sampled as part of the current study, only scalded tripe was requested. The large variation in counts of tripe may be a result of participating processors not adhering to the sampling specification, inadequate time-temperature during scalding, or cross contamination after scalding. Tongue samples in the present study had the highest APC. Rinsing, sufficient for the visual removal of saliva and other materials, may not be sufficient for the removal of bacteria, particularly given the inherent roughness of the tongue which may protect bacteria from being rinsed off the surface.

Other offal, such as head meat and cheek meat, can be heavily contaminated prior to and during processing as the mouths of animals can contain large numbers of bacteria [20,21]. The head is commonly washed and flushed, causing the spread of bacteria from the mouth. The success of the rinsing process is judged by the visual removal of saliva and other material, not a microbiological specification.

While there was a significant difference in the APC on frozen and chilled samples in the current study, the magnitude of the overall difference was considered not biologically significant. Similar results were observed for *E. coli*, in chilled and frozen samples, which contrasts with published data suggesting that freezing can result in decreases in *E. coli* numbers ranging between 0.5 and 0.9 log units compared to chilling alone [8], though the loss in viability may depend on the rate of freezing [22]. The magnitude of the observed reduction during freezing in the current study varied between offal type.

It has been suggested that the inadequate control of cooling can lead to poor microbiological quality of offal [2,20]. Sheridan and Lynch [23] observed significant bacterial growth on beef hearts, livers, and kidneys after overnight refrigeration. While no pre-cooling samples were examined in the current study, final counts on organ offal were similar to those reported on meat, suggesting that cooling, at least for these offal types, is adequately controlled at these processing establishments. The refrigeration of offal for export from Australia, such as muscle meat, is required to comply with the Refrigeration Index criteria, designed to ensure adequate refrigeration [24].

There may be significant differences in processing methods for offal between establishments, such as different methods of transferring single and/or multiple offal from one part of the establishment to another for further processing and packing. Offal can be stored in tubs, dolleys, trays, or buckets before being physically transferred to another room or dropped via a designated chute. Physical inspection, including incision and palpation, which is a regulatory requirement, has been shown to contribute to transferring bacteria from one offal product to another [25]. The transfer of bacteria can occur from contact with viscera while sorting offal on the viscera table, and cross-contamination during washing or rinsing processes. Delmore [20] identified product transfer chutes, holding times before refrigeration, equipment sanitation, packing, and refrigeration as areas where potential improvements in microbiological quality can be made.

The microbiological quality of offal has previously been considered to have a lower standard than muscle meat, although there are few comprehensive studies on the subject. The APC on six beef offal types prior to chilling at two New Zealand plants was reported to be in the range of 3.3 Log_10_ CFU/g (kidneys) to 4.2 Log_10_ CFU/g (tripe), with an *E. coli* prevalence ranging from 30% (kidneys) to 100% (tripe) [26]. Delmore et al. [20] examined 17 offal types in the United States of America finding APC in the range of 3.0–5.2 Log_10_ CFU/g. Im et al. [9] reported average APCs on beef offal processed in Korea ranging from 4.02 Log_10_ CFU/g on hearts to 5.55 Log_10_ CFU/g on tripe. An Australian study conducted across a smaller number of processors but a wider range of offals [27] produced data similar to this study, indicating a lack of selection bias. The microbiological quality of offal reported in this study compares very favourably with results from previously published studies for carcases and cuts of meat. However, laboratory methods used in each study differed and therefore comparisons are not easily made. In general, organ offal are processed separately from non-organ offal, which helps to reduce the level of contamination on these offal types.

This study has set a contemporary baseline for the hygienic quality of offal processed in Australian export establishments and proposed a rationale for the levels of contamination observed. While the quality of some offal approach that of muscle meat, the location of the offal in the animal and the processing steps involved affect the observed hygienic quality. Further attention to hygiene may not result in an improved safety outcome considering most offal are cooked prior to consumption; offal which are consumed raw or undercooked are high risk even for the general population [28].

## Figures and Tables

**Figure 1 foods-11-03007-f001:**
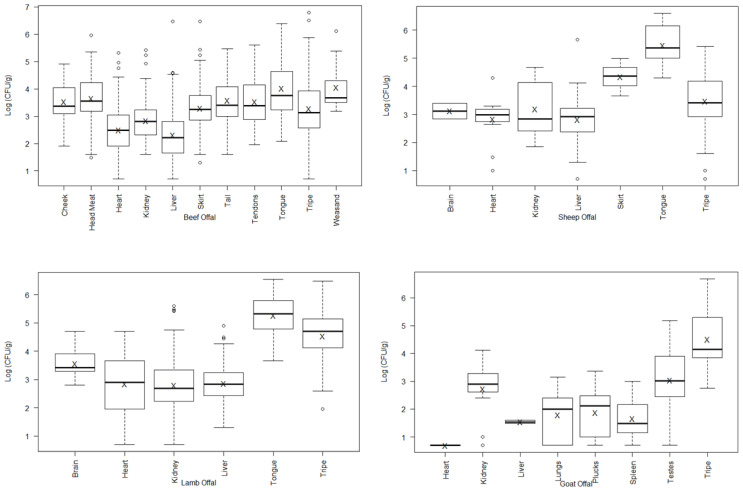
Box plots of the log_10_ APC on offal from beef, sheep, lamb, and goat (both chilled and frozen). The box encompasses data between the 25th and 75th percentile, with the mean indicated by ‘X’ and median by ‘− ’. The statistically expected values are indicated by the whiskers above and below the box, and outlier values by ‘o’. For some sheep and goat offal types, only a small number of samples were analysed and omitted from this figure.

**Table 1 foods-11-03007-t001:** Number of offal samples analysed for each species.

	Species			
Offal Type	Beef	Sheep	Lamb	Goat
Brain	- ^a^	2	25	-
Cheek	75	-	-	-
Head Meat	89	-	-	-
Heart	101	13	26	3
Kidney	60	20	133	23
Liver	107	32	110	2
Lungs	-	-	-	13
Pluck	-	-	-	14
Skirt	115	3	-	-
Spleen	-	-	-	16
Tail	102	-	-	-
Tendons	54	-	-	-
Testes	-	-	-	22
Tongue	103	10	107	-
Tripe	156	80	85	42
Weasand	13	-	-	-

^a^, “-”: indicates that this offal type was not available for this species or production volumes were too low to warrant inclusion in the study.

**Table 2 foods-11-03007-t002:** Prevalence and level of indicator bacteria quantified on beef, sheep, lamb, and goat offal samples. Prevalence values in columns for each indicator with the same letter are not significantly different. Data presented are for both chilled and frozen samples combined.

Indicator/ Species	Prevalence				Average log_10_ CFU/g ^1^
Coliforms					
Beef	30.5%	a			1.54 ± 0.64
Sheep	38.1%	a, b			1.65 ± 0.63
Lamb	37.4%	b			1.67 ± 0.71
Goat	43.0%	b			2.05 ± 0.78
*E. coli*					
Beef	15.4%		a		1.42 ± 0.63
Sheep	28.1%		b		1.52 ± 0.57
Lamb	17.5%		a		1.44 ± 0.59
Goat	39.3%		c		1.82 ± 0.62
APC					
Beef	99.0%			a	3.25 ± 1.06
Sheep	98.8%			a	3.38 ± 1.09
Lamb	99.0%			a	3.70 ± 1.34
Goat	85.9%			b	2.97 ± 1.53

^1^ Average log_10_ APC data were censored by assigning a value of ½ the LOD (10 CFU/g) to the few samples in which this occurred. All other counts are the average of positive samples only. Values in columns for each indicator with the same letter are not significantly different.

**Table 3 foods-11-03007-t003:** Microbiological loads for different offal types. Log_10_ APCs with the same letter in the same column are not significantly different (*p* < 0.05). Some sheep and goat offal types were combined under the heading ‘Other’ (brain, heart, skirt, and tongue for sheep and heart, liver, and spleen for goat). Lung and pluck data for goat were combined as ‘pluck’.

Offal Type	log_10_ APC ^1^							
	Beef		Lamb		Sheep		Goat	
Liver	2.30 ± 1.01	b	2.86 ± 0.62	a	2.81 ± 0.90	b		
Heart	2.50 ± 0.91	b, c	2.83 ± 1.02	a				
Kidney	2.83 ± 0.82	c	2.80 ± 1.01	a	3.18 ± 0.93	a, b	2.73 ± 1.02	a
Tripe	3.28 ± 1.14	a	4.54 ± 0.95		3.46 ± 0.97	a	4.51 ± 1.03	
Skirt	3.29 ± 0.83	a						
Cheek	3.53 ± 0.62	a						
Tendons	3.53 ± 0.95	a						
Tail	3.57 ± 0.78	a						
Head Meat	3.65 ± 0.81	a, d						
Tongue	4.01 ± 1.05	d	5.26 ± 0.74					
Weasand	4.06 ± 0.84	a, d						
Brain			3.55 ± 0.40					
Pluck							1.83 ± 0.91	b
Testes							3.05 ± 1.26	a
Other					3.95 ± 1.42	a	1.51 ± 0.73	b

^1^ Average log_10_ APC data were censored by assigning a value of ½ the LOD (10 CFU/g) to the few samples in which this occurred. Values in columns for each indicator with the same letter are not significantly different.

**Table 4 foods-11-03007-t004:** Faecal indicators (average log_10_ *E. coli* prevalence and count) for different offal types. For some sheep and goat offal types, only a small number of samples was analysed and combined under the heading ‘Other’ (brain, heart, skirt, and tongue for sheep and heart, liver, and spleen for goat). Lung and pluck data for goat were combined under the category ‘pluck’.

Offal Type	Beef		Lamb		Sheep		Goat	
	Prevalence	Count ^1^	Prevalence	Count ^1^	Prevalence	Count ^1^	Prevalence	Count ^1^
Liver	9.3%	1.69 ± 0.87	17.3%	1.31 ± 0.46	9.4%	1.91 ±0.82		
Heart	5.9%	1.61 ± 0.99	0%	- ^2^				
Kidney	8.3%	1.45 ± 0.62	4.5%	1.40 ± 0.44	45%	1.36 ± 0.51	26.1%	1.35 ± 0.33
Tripe	10.9%	1.77 ± 0.99	29.4%	1.52 ± 0.73	30%	1.57 ± 0.61	76.2%	1.87 ± 0.61
Skirt	19.1%	1.39 ± 0.70						
Cheek	17.3%	1.23 ± 0.36						
Tendons	9.3%	1.37 ± 0.67						
Tail	20.6%	1.26 ± 0.28						
Head Meat	32.6%	1.42 ± 0.52						
Tongue	15.5%	1.24 ± 0.34	29%	1.48 ± 0.59				
Weasand	46.2%	1.41 ± 0.40						
Brain			16%	1.27 ± 0.32				
Pluck							3.7%	2.76 ^3^
Testes							60.9%	1.85 ± 0.66
Other					32.1%	1.42 ± 0.43	0%	- ^2^

^1^ Average log_10_
*E. coli* is for positive samples only, ^2^ not detected, ^3^ no SD as only one sample positive.

**Table 5 foods-11-03007-t005:** APC on offal samples (all species combined) collected chilled or frozen. Only counts for offal types where more than 30 samples were analysed from both frozen and chilled samples.

Offal Type	Average log_10_ CFU/g ^1^		
	Chilled	Frozen	Difference
Tripe	4.49 ± 0.71	3.69 ± 1.20	0.80
Tongue	5.05 ± 1.09	4.46 ± 1.04	0.58
Kidney	3.19 ± 0.94	2.63 ± 0.91	0.56
Liver	2.90 ± 0.77	2.47 ± 0.91	0.43
Skirt	3.29 ± 1.03	3.33 ± 0.73	−0.05

^1^ Average log_10_ APC data were censored by assigning a value of ½ the LOD (10 CFU/g) to the few samples in which this occurred.

## Data Availability

The data presented in this study are available on request from the corresponding author. The data are not publicly available due to privacy obligations to the participating processing establishments.

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
