# Peer review of "Microbiological Quality of Red Meat Offal Produced at Australian Export Establishments"

_foods, 2022, doi:10.3390/foods11193007_

Round 1
Reviewer 1 Report
GENERAL COMMENTS
Dear authors, congratulations on your manuscript,
I have some major questions.
Kind Regards,
Reviewer,
SPECIFIC COMMENTS
Materials and Methods: All ISO and AOAC must be properly cited throughout the text and properly described in the references.
Line 99 - Delete. This information is already on line 59.
Tables 2,3, and 4: Any average should be accompanied by a standard deviation. Why don't the authors present mean ± SD?
Table 2: The prevalence of Coliforms, E. coli and APC were statistically compared to each other??? As there is a continuity of the letters a,b,c and d, it gives that feeling, but it doesn't make sense for it to be so. I think that for E. coli and APC the letter a should be used first and then the others, if appropriate.
The authors note "Prevalence values in columns, for each indicator, with the same letter are not significantly different". For example, for E. coli we know that there were no differences between prevalence obtained between beef and lamb, but we do not know for goat and sheep... another thing i don't understand, how come 30.5% (a), 38.1% (a,b) and 37.4% which is less than 38.1% is b?
Lines 111 and 112: The authors state "There was a lower prevalence of coliforms in bovine offal compared to offal from other species", but the % value obtained, in statistical terms, for sheep is similar to that of beef?
Line 113: Figure???
Lines 124 and 125: I do not understand where the values in table 3 are shown?
Lines 126 and 127: Why is table 3 bold, in line 124, and table 4 and table 2 not?
Line 130: Frozen and chilled? but how do we identify chilled and frozen samples in Table 3?
Lines 133 to 136: I do not understand the results (values) presented...
Line 139: add p<0.05
To be able to analyse the discussion I need to see the above questions clarified.
Author Response
SPECIFIC COMMENTS
Materials and Methods: All ISO and AOAC must be properly cited throughout the text and properly described in the references.
Response: Complete citations for ISO and AOAC documents have now been added to the text and the reference list
Line 99 - Delete. This information is already on line 59.
Response: deleted
Tables 2,3, and 4: Any average should be accompanied by a standard deviation. Why don't the authors present mean ± SD?
Response: standard deviations for the data have now been added to all tables, and the method for calculation has been added to the statistical methods
Table 2: The prevalence of Coliforms, E. coli and APC were statistically compared to each other??? As there is a continuity of the letters a,b,c and d, it gives that feeling, but it doesn't make sense for it to be so. I think that for E. coli and APC the letter a should be used first and then the others, if appropriate.
Response: We agree that this could be confusing, though we believe our explanation of how to read the table was sufficient. We have chosen to indicate statistic differences for each of coliforms, E. coli and APC in different columns so that it is clear that they are not to be compared. Letters indicating difference have been added to every line of the table.
The authors note "Prevalence values in columns, for each indicator, with the same letter are not significantly different". For example, for E. coli we know that there were no differences between prevalence obtained between beef and lamb, but we do not know for goat and sheep... another thing i don't understand, how come 30.5% (a), 38.1% (a,b) and 37.4% which is less than 38.1% is b?
Response: We have checked our calculations and assignment of letters to indicate significant difference and they are correct. The reason for the apparent anomaly is because the chi-squared statistic depends on the number of samples, which we not equal in this case.
Lines 111 and 112: The authors state "There was a lower prevalence of coliforms in bovine offal compared to offal from other species", but the % value obtained, in statistical terms, for sheep is similar to that of beef?
Response: The difference between bovine and ovine offal has been explained now as line 112 “: There was lower prevalence of coliforms on beef offal compared to offal from other species (lamb and sheep data combined).”
Line 113: Figure???
Response: For some reason the figure number was not reproduced in the review version of the document, though it remains in the other versions. The reference is to Figure 1.
Lines 124 and 125: I do not understand where the values in table 3 are shown?
Response: Tables have now been moved so that they are all before the text of the discussion.
Lines 126 and 127: Why is table 3 bold, in line 124, and table 4 and table 2 not?
Response: There seems to have been some problem with transmission of the manuscript to the reviewer/ The version from the editorial office does not have bold text.
Line 130: Frozen and chilled? but how do we identify chilled and frozen samples in Table 3?
Response: we have now added Table 5 which provides APC data for the five offals for which a sufficient large number of samples were obtained to allow statistical analysis.
Lines 133 to 136: I do not understand the results (values) presented...
Response: We could not locate values at these lines, so, unfortunately, we cannot respond to the reviewer’s comments. Hopefully, they have been answered by our other comments and modffications.
Line 139: add p<0.05
Response: It isn't clear where this addition can be made in line 139 or those around it. We are here presenting a summative statement of our work, which we further elucidate in the following paragraphs. It is possible that your review copy and our copy of the manuscript have different line numbering.
Reviewer 2 Report
A very interesting and highly informative manuscript on the microbiological quality of red meat offal produced at Australian export establishments. The manuscript is very well written, the materials and methods are suitable for such type of study and the results are clearly presented and thoroughly discussed. Only a few minor suggestions can be offered:
l. 113. it should read ‘(Figure 1)’
l. 124. the sentence seems truncated, it seems that a part of it is missing, please verify
Author Response
A very interesting and highly informative manuscript on the microbiological quality of red meat offal produced at Australian export establishments. The manuscript is very well written, the materials and methods are suitable for such type of study and the results are clearly presented and thoroughly discussed. Only a few minor suggestions can be offered:
- 113. it should read ‘(Figure 1)’
response: For some reason the figure number was not reproduced in the review version of the document, though it remains in the other versions
- 124. the sentence seems truncated, it seems that a part of it is missing, please verify
response:
response: Thank you for pointing this out. The sentence was not well-constructed and has been re-written so that its meaning is clearer.
Reviewer 3 Report
Procedures.
2.1. Please indicate locations of the establishments.
2.2. Table 1 must be moved here.
2.3. Please describe specific tests performed for analysis.
2.4. Please delete altogether.
Results.
Tables 3 and 4 and Figure 1 are badly placed in the manuscript.
Discussion.
We propose that the hygienic quality of red meat offal can be 146 understood by considering, the anatomical site from which they are harvested, the usual 147 bacterial levels found at that site, the difficulty in hygienically … -> Please rephrase, the statement is reasonable, you do not need to just propose it.
The discussion is a bit on the verbose side, it can easily reduced in length by 20%-25% and the manuscript will be more reader-friendly.
Overall. The changes suggested should be implemented.
The manuscript is useful and can advance to the next stage. However, it should be shortened as indicated and should be resubmitted as communication.
Author Response
Procedures.
2.1. Please indicate locations of the establishments.
Response: The distribution of the processing establishments across Australia have been indicated by the additional text: “distributed across five out of six of the states of Australia,”’
2.2. Table 1 must be moved here.
Response: the table has been moved as requested.
2.3. Please describe specific tests performed for analysis.
Response: the statistical tests performed have now been indicated.
2.4. Please delete altogether.
Response: This manuscript makes a significant contribution to the literature by providing an explanation (in the Discussion section) for the microbiological results obtained. We therefore, would like to acknowledge, that this was possible due to the consultations held, and retain this section of the method.
Results.
Tables 3 and 4 and Figure 1 are badly placed in the manuscript.
Response: We found it difficult to place the Tables and Figures. The Tables and Figures have been moved so that they all appear prior to the Discussion text.
Discussion.
We propose that the hygienic quality of red meat offal can be 146 understood by considering, the anatomical site from which they are harvested, the usual 147 bacterial levels found at that site, the difficulty in hygienically … -> Please rephrase, the statement is reasonable, you do not need to just propose it.
Response: Thank you for your suggestion and support. We have modified the sentence.
The discussion is a bit on the verbose side, it can easily reduced in length by 20%-25% and the manuscript will be more reader-friendly.
Response: We have removed some of the details from the other work cited, which has reduced the length and made it easier to read. We believe that one of the significant contributions of this work has been to provide reasons why offal may have the microbiological counts observed, and this is incorporated into the discussion. We think that the discussion is now appropriate.
Overall. The changes suggested should be implemented.
The manuscript is useful and can advance to the next stage. However, it should be shortened as indicated and should be resubmitted as communication.
Response: We have agreed with the editorial office that the manuscript be considered a ‘’communication’’
Reviewer 4 Report
This paper focuses on the hygienic quality of beef, sheep, lamb and goat offal produced in different Australian export establishments.
In my opinion, the experimental design is quite well described and the paper provides an interesting food safety issue.
However I found the Results section difficult to read and I suggest the authors to revise it.
In particular:
- Figure 1, Table 3 and Table 4: why are the mentioned in the Results section but they are included in the Discussion section?
- Line 113: “The average log10 APC varied between offal type within species (Figure )”, the number of the figure is not indicated.
- Line 114: I don’t understand the bracket. Moreover, the sentence (line 114-122) is the same in the Discussion section (line 177-186), I think that it is not necessary to repeat it twice.
- Line 124-126: Please check this part, I think it is necessary to rephrase it.
- Line 187 (Table 3): “Microbiological loads for different beef offal types”, why only beef? In the Table, I can see all the species.
Author Response
Comments and Suggestions for Authors
This paper focuses on the hygienic quality of beef, sheep, lamb and goat offal produced in different Australian export establishments.
In my opinion, the experimental design is quite well described and the paper provides an interesting food safety issue.
However I found the Results section difficult to read and I suggest the authors to revise it.
In particular:
- Figure 1, Table 3 and Table 4: why are the mentioned in the Results section but they are included in the Discussion section?
Response: We found it difficult to place the Tables and Figures. The Tables and Figures have been moved so that they all appear prior to the Discussion text.
- Line 113: “The average log10 APC varied between offal type within species (Figure )”, the number of the figure is not indicated.
Response: We are sorry that this occurred. For some reason the figure number was not reproduced in the review version of the document, though it remains in the other versions
- Line 114: I don’t understand the bracket. Moreover, the sentence (line 114-122) is the same in the Discussion section (line 177-186), I think that it is not necessary to repeat it twice.
Response: We have added units to the numbers in the bracket, which makes our meaning clearer. We are presenting here, in words, an interpretation of the data in the table at lines 177-186 and an explanation of our inferences. To be clear, we have repeated some of the data in the table.
- Line 124-126: Please check this part, I think it is necessary to rephrase it.
Response: the sentence was not well-constructed and has been re-written so that its meaning is clearer.
- Line 187 (Table 3): “Microbiological loads for different beef offal types”, why
only beef? In the Table, I can see all the species.
Response: Thank you for pointing out this error, which has been corrected
Round 2
Reviewer 1 Report
GENERAL COMMENTS
Dear authors, congratulations on your manuscript,
I have some minor questions.
Kind Regards,
Reviewer,
SPECIFIC COMMENTS
Line 127 - Figure???
Line 140 - delete )
References: in the references I don't think it makes sense to put anonymous in ISO. I suggest you remove the anonymous
Author Response
Line 127 - Figure???
I cannot find a reference to a figure in this line number of the resubmitted or the returned manuscript. I cannot find the word 'figure'' failing to refer to Figure 1.
On the assumption that this comment refers to the caption for Figure 1, I have made it clear that some data have been omitted from this figure, to ensure that there is no confusion.
Line 140 - delete )
Thank you. I searched the manuscript and found a stray ) but it was in line 237 of the current version of the manuscript.
References: in the references I don't think it makes sense to put anonymous in ISO. I suggest you remove the anonymous
Thank you for this suggestion, which we will follow. Some journals require an author for every document cited.
Reviewer 3 Report
Section 2.1. The precise locations must be indicated, preferably on a map. Abstract expression as the one inserted, raise suspicions......
Section 2.4. This needs to be transferred to the specific section for acknowledgements. It can be retained in M&M, if there was a structured questionnaire and a methodological protocol for these discussions, in which case these two must also have presented for reviewing.
Author Response
Section 2.1. The precise locations must be indicated, preferably on a map. Abstract expression as the one inserted, raise suspicions......
The approach taken for this work has been used in numerous surveys, peer-reviewed and published in reputable journals. All of the establishments were export registered, and supervised by the same competent authority. The purpose of sampling from multiple establishments was to provide a fair assessment of the hygiene of offals from Australian export establishments, and we believe that the design has achieved that. Additional text provides the reader with a deeper understanding of the rationale for sampling.
We cannot respond to the reviewers suspicions without them being explicitly stated.
Section 2.4. This needs to be transferred to the specific section for acknowledgements. It can be retained in M&M, if there was a structured questionnaire and a methodological protocol for these discussions, in which case these two must also have presented for reviewing.
response: As suggested, this part of our method has been deleted from M&M and relocated as an acknowledgement.